# Autophagy: A Potential Therapeutic Target of Polyphenols in Hepatocellular Carcinoma

**DOI:** 10.3390/cancers12030562

**Published:** 2020-02-29

**Authors:** Chandramohan Kiruthiga, Kasi Pandima Devi, Seyed M. Nabavi, Anupam Bishayee

**Affiliations:** 1Department of Biotechnology, Alagappa University (Science Campus), Karaikudi 630 003, Tamil Nadu, India; kiruthipugal2006@gmail.com; 2Applied Biotechnology Research Center, Baqiyatallah University of Medical Sciences, Tehran 1435916471, Iran; nabavi208@gmail.com; 3Lake Erie College of Osteopathic Medicine, Bradenton, FL 34211, USA

**Keywords:** hepatocellular carcinoma, polyphenols, autophagy, anticancer effects, therapy

## Abstract

Autophagy is a conserved biological phenomenon that maintains cellular homeostasis through the clearing of damaged cellular components under cellular stress and offers the cell building blocks for cellular survival. Aberrations in autophagy subsidize to various human pathologies, such as dementia, cardiovascular diseases, leishmaniosis, influenza, hepatic diseases, and cancer, including hepatocellular carcinoma (HCC). HCC is the fifth common mortal type of liver cancer globally, with an inhomogeneous topographical distribution and highest incidence tripled in men than women. Existing treatment procedures with liver cancer patients result in variable success rates and poor prognosis due to their drug resistance and toxicity. One of the pathophysiological mechanisms that are targeted during the development of anti-liver cancer drugs is autophagy. Generally, overactivated autophagy may lead to a non-apoptotic form of programmed cell death (PCD) or autophagic cell death or type II PCD. Emerging evidence suggests that manipulation of autophagy could induce type II PCD in cancer cells, acting as a potential tumor suppressor. Hence, altering autophagic signaling offers new hope for the development of novel drugs for the therapy of resistant cancer cells. Natural polyphenolic compounds, including flavonoids and non-flavonoids, execute their anticarcinogenic mechanism through upregulating tumor suppressors and autophagy by modulating canonical (Beclin-1-dependent) and non-canonical (Beclin-1-independent) signaling pathways. Additionally, there is evidence signifying that plant polyphenols target angiogenesis and metastasis in HCC via interference with multiple intracellular signals and decrease the risk against HCC. The current review offers a comprehensive understanding of how natural polyphenolic compounds exhibit their anti-HCC effects through regulation of autophagy, the non-apoptotic mode of cell death.

## 1. Introduction

Hepatocellular carcinoma (HCC) is the third common lethal human cancer globally, with higher incidence and mortality rates in men than women. The incidence varies with an inhomogeneous topographical distribution, higher in developing areas; males are mainly affected, with a ratio of 2:3 female/male. According to the International Agency for Research on Cancer, which is associated with the World Health Organization, HCC accounts for more than 782,000 deaths each year. HCC has been a growing worldwide problem with a high mortality rate and its poor prognosis is due to complicated pathogenesis [1,2]. Current options for treatment of HCC are chemotherapy, radiotherapy, and surgery. However, each therapy has serious adverse effects and a satisfactory therapeutic effect achieved by combined therapy increases the side effects. Therefore, it is essential to find a complementary option with better efficacy and fewer side effects for HCC. Liver malignancies developed by a number of genetic and epigenetic mechanisms trigger resistance to cell death and cause upregulation of survival mechanisms. Autophagy is a conserved biological pathway that maintains cellular homeostasis. Liver cells induce autophagy in order to provide functional biosynthesis, and also for regulating and recycling damaged organelles, such as mitochondria [3]. Recent data has shown that autophagy is involved in liver pathogenesis and tumor development. Autophagic aberrations cause oxidative stress, contributing to gene alterations, and could transform the cells to change their metabolism with the development of malignant cells. Autophagy, by contrast, acts as an anticancer mechanism, inhibiting the malignant transformation of normal cells into cancer cells [4]. In comparison, autophagy is also implicated in different stages of cancer development and metastasis. The survival of fast-growing tumors is in particular correlated with their autophagous activity. There is also a large collection of articles concerning autophagy in drug resistance [5]. In HCC, autophagy may involve both tumor surveillance and invasiveness, indicating a dual role and stage-dependent function in cancer. Since several cancer drugs were identified, which directly or indirectly manipulate autophagy and would help cell-type-directed strategies, it is interesting to explore autophagy as a target for HCC treatment [3]. In recent days, natural medicines have been accepted as effective anticancer drugs and neoadjuvants to improve the effectiveness and reduce the side effects of chemotherapy. Several anticancer drugs, such as vinblastine, teniposide, camptothecin, docetaxel, etoposide, and paclitaxel, have been either identified in natural sources or synthesized by modification of molecules having natural origins [6,7]. Numerous natural products, including phytochemicals of dietary and non-dietary origin, have been shown to modulate various signaling pathways linked to hepatocarcinogenesis in vitro and in vivo, which could be valuable for the prevention and treatment of HCC [8,9,10,11,12,13,14,15,16].

Flavonoids and non-flavonoids belong to polyphenols, which are the most common group of phytocompounds enriched in plants and available to humans through diet. Polyphenolic compounds show a wide range of biological functions, including antioxidant, antiallergic, anti-inflammatory, antimutagenic, anticarcinogenic, and antiproliferative effects [17,18,19,20]. In addition, they exhibit co-chemotherapeutic effects by various mechanisms, such as cell cycle alteration, antiproliferation, apoptosis, autophagy, and modulation of various cell signaling pathways [21]. Previous studies have demonstrated that polyphenolic compounds, such as genistein, quercetin, resveratrol, curcumin, apigenin, luteolin, casticin, chrysin, and 8-bromo-7-methoxychrysin, as well as their synthetic analogues have shown tumor growth-inhibiting effects [22,23]. Recently, it has been revealed that quercetin, resveratrol, curcumin, chrysin, casticin, and 8-bromo-7-methoxychrysin inhibit the growth of tumor stem cells [24].

Various natural compounds, including polyphenols, have recently been known to exhibit anticancer effects via autophagy modulation [25,26,27,28]. For example, by controlling autophagy, plant flavonoid wogonin has been shown to cause cancer cell death [29] and curcumin can induce autophagy- and apoptosis-mediated cell death in chronic myeloid leukemia cells by lowering the B-cell lymphoma 2 (Bcl-2) protein [30]. Quercetin induces widespread autophagy in colon epithelial cancer cells, leading to cell cycle arrest and initiation of apoptosis [31]. Further, polyphenols execute their anti-HCC effects through autophagy by interfering with canonical (Beclin-1-dependent) and non-canonical (Beclin-1-independent) pathways as well as various other signaling, including epidermal growth factor receptor (EGFR)/tyrosine-protein kinase Met (c-Met) signaling pathway, nuclear factor κ-light-chain-enhancer of activated B cells (NF-κB)-related pathway, Janus kinase/signal transducer and activator of transcription pathway, heat shock protein-related pathway, mitogen-activated protein kinase (MAPK)/c-Jun N-terminal kinases (JNK) pathway, phosphoinositide-3-kinase (PI3K)-phosphatase and tensin homolog-serine/threonine-protein kinase (PTEN)-mammalian target of rapamycin (mTOR), and transforming growth factor-β (TGF-β) pathway. In addition, evidence indicates that polyphenols target angiogenesis and metastasis in HCC through regulation of multiple intracellular signals and reducing the risk of HCC [32]. However, there is a scarcity of comprehensive work that exclusively analyzes the current evidence on the autophagy-modulating role of polyphenols in HCC. Accordingly, the current review offers a comprehensive and in-depth understanding of how natural polyphenols exhibit their anti-HCC effect through non-apoptotic cell death, such as autophagy. Based on the analysis of the available literature presented here, polyphenols could be considered as potent autophagy modulators for HCC therapy. Although numerous in vitro and in vivo studies have reported the beneficial roles of polyphenols in treating HCC [8,9,10,11,12,13,14,15,16], there are hardly any clinical trials to realize the full potential of preclinical studies. For this purpose, it is worthwhile to explore polyphenols alone or in combination with approved drugs for the treatment of human HCC.

## 2. Autophagy: An Overview

Autophagy is a preserved biological process that monitors cellular homeostasis between synthesis and unwanted cellular components’ degradation [33]. Autophagy may be either specific or non-specific in the bulk removal of misfolded proteins, aggregated proteins, and impaired organelles’ degradation, such as mitochondria, endoplasmic reticulum, and peroxisomes, in addition to the elimination of intracellular pathogens [34]. Autophagy is generally thought to be a mechanism for survival; however, its continuous activation may lead to non-apoptosis-mediated cell death.

The known classes of autophagy are micro-autophagy, macro-autophagy, and chaperone-mediated autophagy (CMA), all of which are common in promoting proteolytic elimination of cytosolic components at the lysosome. Macro-autophagy is characterized by the development of a double-membranous vesicle termed autophagosomes that swallow a useless cytosolic portion. Subsequently, it delivers to the lysosome, where engulfed components are fragmented and brought back to the cytosol [34,35]. In comparison, the lysosome itself directly absorbs the cytosolic components by the lysosomal membrane invagination in micro-autophagy. In CMA, chaperone proteins form a complex with the targeted proteins, which are identified by the lysosomal membrane receptor lysosomal-associated 2A membrane and translocated across the lysosomal membrane, resulting in their degradation [36]. Recent research focuses specifically on macro-autophagy (referred to as “autophagy”), its cellular and molecular aspects, and how it can be exploited in anticancer properties [37,38].

### 2.1. Conservative Modes of Cell Death

#### 2.1.1. Programmed Cell Death

The three conservative modes of cell death, namely apoptosis, autophagy, and necrosis, exhibit diverse morphological features by activating different signaling pathways (Figure 1). Apoptosis is evolutionally conserved, well studied as caspase-mediated programmed cell death (PCD) or type I PCD [39]. Apoptotic-mediated PCD is characterized by various molecular signals, biochemical changes, and morphological alterations, including the phosphatidyl-l-serine access to the outer plasma membrane (early apoptosis), cell shrinkage, plasma membrane blebbing, chromosome condensation, nuclear fragmentation (late apoptosis), and development of apoptotic bodies [40]. Apoptosis, a critical process in oncology by which cancer cells can be eliminated, is exhibited by different molecular mechanisms, such as the exogenous pathway, intrinsic pathway, and endoplasmic reticulum stress. Interestingly, HCCs are resistant to apoptosis induced by death receptors, such as Fas ligand (FasL) or TNF-related apoptosis-inducing ligand (TRAIL). Whereas the expression of some pro-apoptotic genes is downregulated and HCC dysregulates the balance between death and survival mainly due to upregulation of the anti-apoptotic signaling mechanism. In several HCC cells, the overactivation of the Janus tyrosine kinase (JAK)/signal transducers and activators of transcription (STAT), PI3K/protein kinase B (Akt), and Ras/extracellular signal-regulated kinase (ERK) pathways confer resistance to apoptotic stimuli on them [41].

Necroptosis is referred to as programmed necrosis or inflammatory cell death. Necroptosis is a regulated inflammatory mode of cell death caused by a physical trauma involving a caspase-independent signaling pathway [42] (Figure 1). Accumulated studies have been conducted on tumor necrosis factorα (TNFα), receptor interacting protein kinase 3 (RIPK3), and caspase-8 to determine the molecular mechanism of necroptosis. Necroptosis may be initiated by TNF superfamily receptors, T cell receptors, toll-like receptors (TLR3 and TLR4), and interferon receptors. TNF receptor superfamily-mediated necroptosis is well characterized.Until recently, the role of necroptosis in HCC progression and its regulatory mechanisms were hardly investigated.

#### 2.1.2. Autophagy

Autophagic cell death is referred to as type II cell death. Aberrations or genetic inhibition of the autophagy signaling pathway will lead to cell death, and it is identified principally by the presence of autophagosomes (Figure 1). Autophagy is a multifaceted process that involves autophagy-related proteins (Atg), ubiquitin-like conjugation systems (Atg12 and LC3 systems), and these systems make autophagy regulators, such as Atg8-PE and Atg5-Atg12-Atg16. These autophagic regulators may determine the nucleation, expansion, autophagosome formation, and lead to lysosome fusion. Formation of the ULK1 complex is important for the phagophore assembly site consisting of ULK1, Atg13, RB1-inducible coil protein 1 (FIP200), and Atg101. mTORC1 regulates ULK1/2 complex, which is downregulated by AMPK and upregulated by PI3K/Akt. The PI3K/Akt pathway could be activated by growth factors through RTKs. Further, ULK1/2 complex triggers nucleation by phosphorylating class III PI3K complex I, Beclin-1, activating molecule in Beclin-1-regulated autophagy protein 1 (AMBRA1), Atg14, and general vesicular transport factor (p115), which pushes the insulation membrane to nucleation. In addition, the other complex includes Atg1 serine/threonine kinase. The kinase activity of Atg1 requires the function of two other autophaghic proteins, such as Atg8 and Atg17. Further, Atg1 was found to be associated with the Atg8 orthologues, Golgi-associated ATPase enhancer of 16 KDa (GATE-16), microtubule-associated protein light chain 3 (LC3), and G-amino butyric acid type A receptor-associated protein (GABARAP). The Atg8 protein mediates autophagosome formation, wherein the phosphatidyl ethanolamine is covalently bound with Atg8, and the activated Atg8 and vesicle membrane protein 1 may be involved in lipid transportation to the membrane nucleation. Subsequently, the autophagosome and lysosome fusion is mediated by SNARE-like protein, including syntaxin 17, synaptosomal-associated protein 29, and vesicle-associated membrane protein 8, and facilitates lysosomal degradation to hydrolyze all the unwanted proteins, lipids, nucleic acid, and damaged organelles [34,43,44]. The signaling pathway of autophagy-mediated cell death or the caspase-independent mechanism has not been well investigated. However, several findings reveal that autophagic cell death may be due to changes in the morphologic characteristics, such as extensive autophagosomal/autolysosomal formation and Atg-8/LC3 translocation to autophagic vesicles. Further, some reports showed that autophagic cell death through caspase inhibition by selective degradation of catalase further leads to the production of reactive oxygen species (ROS) that kill the cell [45]. The precise role of autophagy has not been fully determined in HCC and is contentious. Hence, in-depth research is needed to understand the role of autophagy in HCC growth.

### 2.2. Autophagy-Regulating Signaling Pathways

Autophagy is triggered during food deprivation, hyperthermia, hypoxia, oxidative response, and pathogen infection for balancing the energy sources in critical development times [46]. Obviously, autophagy responds to nutrition balancing for cell viability and metabolic homeostasis. The accumulation of evidence facilitates the understanding of the molecular mechanisms that play a role in the regulation of autophagy. Nevertheless, complete knowledge behind how the signaling cascades precisely perform in autophagy regulation is still not clear. Subsequently, autophagy is linked to pathological and physiological conditions directed by a complicated web of signaling cascades that cross-talk with each other to create a specific response.

#### 2.2.1. TOR Complex 1 (TORC1) and Ras/Protein Kinase A (PKA) Pathway

In the presence of nutrients, rapamycin inactivates TORC1, which induces autophagy in the yeast cycle [47]. Sufficient evidence suggests that human vacuolar protein sorting 34 is stimulated by extracellular amino acids, resulting in mTOR activation and inhibition of autophagy. Atg13 is hyperphosphorylated by TORC1 in yeast and has alower affinity to Atg1. In addition, TORC1 regulates many effectors involved in the transcription or translation through phosphorylation of certain proteins, some of which are needed for autophagy. In coordination with nutrient transmission pathways, i.e., Ras/cAMP-dependent PKA, TOR controls autophagy induction. Ras/PKA plays an important role in the sensing of glucose from yeast to mammals. Research data supports the suppression of autophagy caused by TOR inhibition by activating the Ras/PKA pathway, indicating that the Ras-PKA pathway decreases autophagy parallel to the TOR pathway [48]. Inhibition of autophagy by Ras/PKA can be mediated by regulating Atg1, known as a PKA substrate phosphorylation [49]. Similar to yeast, mTOR tends to control autophagy in mammals also.

#### 2.2.2. Insulin-Like Growth Factor-1 (IGF-1) Pathway

IGF-1 has been reported to regulate reproductive growth, proliferation, morphogenesis, metabolism, and survival of numerous cell types and tissues [50]. In mammals, the PI3K-Akt-mTOR pathway has been reported to regulate autophagy through the insulin receptor IGF-1 [51]. Additionally, activation of insulin like growth factor 1 receptor (IGF-1R) signaling could be suppressing the autophagic lysosomal pathway [52]. The key components of insulin receptor substrate 1 (IRS1), an adaptor of IGF1R, are involved in IGF-1 signaling pathways. IGF1R phosphorylation induces the binding of IRS1 to IGF1R, which allows IRS1 to stimulate the PI3K pathway [53]. Further, IGF1R-mediated cell survival under stress conditions leads to enhanced activation of autophagy by the suppression of the PI3K-Akt-mTOR signaling pathway [54]. In *Caenorhabditis elegans*, genes of autophagy are important for life-span extension, in which autophagy is reported to have an influence on controlling the cell size converged by the growth modulator signaling pathways, the effects being mediated by insulin/IGF-1 or TGF-β signaling [55]. Understanding the link between the IGF1/IGF1R systems to the autophagy machinery is rather complicated and needs to be explored more in the future.

#### 2.2.3. Damage-Regulated Autophagy Modulator (DRAM) and p53 Pathway

DRAM, autophagy-associated protein5, and/or death-associated protein kinase (DAPK) positively regulate p63 and p73, the members of the p53 family. DAPK is involved in the phosphorylation of Beclin-1 or inhibiting microtubule-associated protein 1B-light chain 3, which induce autophagy. DRAM is a direct mediator of autophagy and apoptosis, which can be upregulated by activated p53 [56]. Apoptosis may also be caused by DRAM and p63/p73. DRAM is also a lysosomal protein that can interact with different autophagosome formation stages [57]. Evidence suggests that tumor suppressor p53 activation can stimulate autophagy [58]. Knockdown of p53 in various cells originated from nematode, mouse, and human can induce autophagy. Enhanced activation of autophagy in p53-deficient cancer cells improved survival under multiple molecular stress conditions. Additionally, autophagy stimulation causes p53 degradation through E3 ubiquitin ligase human double minute 2 homolog [59].

#### 2.2.4. Protein Kinases

Kinases like mTOR, PI3K, and AMPK have strict control on autophagic pathways, whereas kinases like protein kinase C and MAPK may partly influence autophagy by modulating autophagic-associated proteins [60,61]. The specific function of receptor tyrosine kinases’ (RTKs) signaling in autophagy regulation is less well established with respect to the modulation of autophagy-related protein levels [62]. A spectrum of pathological conditions like cancer is linked with the deregulation of different RTKs, such as EGFR, platelet-derived growth factor receptor (PDGFR), hepatocyte growth factor receptor (HGFR), and IGF-1R [62]. The autophagy key players, such as ULK1 and Beclin 1 regulations, are most studied through protein kinases signaling. mTORC1 kinase-dependent signaling complex can prevent autophagy by specific inhibitory phosphorylation of ULK1 and Atg13. Autophagy, on the other hand, can proceed in the absence of ULK1 activity; RTKs can also regulate autophagy by modulating downstream autophagy players [62]. The preceding studies indicate that active and inactive RTKs by controlling Beclin-1 behavior will impose different effects on the autophagy pathway. Activation of RTKs mediates Beclin 1 regulation through direct phosphorylation to suppress the progression of autophagy [63,64,65]. One of the RTKs, EGFR, phosphorylates Beclin 1, resulting in suppression of autophagy and increased tumorigenesis. Complexed with oncoprotein lysosomal-associated transmembrane protein 4B, inactivated EGFR can also modulate autophagy to restore from its inhibitory effects on Beclin-1 and thus initiate autophagy [66].

#### 2.2.5. Forkhead Box Class O (FoxO)

FoxO transcription factors play a vital role in the stress response and trigger the transcription of antioxidant proteins (e.g., superoxide dismutase-2, catalase, peroxiredoxins 3 and 5, ceruloplasmin and selenoprotein P) in the cell [67]. FoxO protein family negatively controls autophagy by inhibiting the transcription process of PC3 cells of prostate cancer [68]. In cultured cardiac myocytes, the elevation of FoxO and sirtuin 1 (Sirt1) leads to an upsurge in the autophagic flux that mediates the autophagosomes’ fusion and lysosomes [69]. Additionally, deacetylation of FoxO protein by Sirt1 is reported to trigger the autophagy in the cardiac tissues during oxidative stress [70]. Acetylated FoxO induced the autophagic process through interaction with Atg7 in autophagosome formation. Apparently, autophagy stimulation depends upon deacetylation or acetylation of FoxO proteins.

### 2.3. Autophagy-Mediated Tumor Suppression

The induction of autophagic activity appears as a robust barrier against malignant transformation involved in cell protection and suppression of tumor initiation, yet the underlying mechanisms are still unclear. The tumor-suppressive role of autophagy during the cellular process through the protective mechanisms comprises of devoid mutagenic ROS accumulation, prevention of DNA damage, suppression of genomic instability, and removal of oncogenic proteins [71]. Whereas this oncosuppressive function of autophagy naturally permits cell survival and prolonged autophagy induction results in autophagic programmed cell death [72]. Conversely, defects in autophagy machinery have been associated with an increased risk of malignancy. Notably, the recent evidence shows that the scaffold protein autophagy and Beclin-1 regulator 1 (AMBRA1) inhibits the protein kinase mTOR and facilitates degradation of proto-oncogene c-Myc, thus regulating cell proliferation, whereas deregulation of c-Myc was shown to have an increased rate of oncogenesis in mice [73]. Consistent with this, PI3K and activated Akt1 are the oncoproteins that act as potent inhibitors of autophagy [74]. Similarly, tumor suppressors are autophagic activators, such as PTEN and liver kinase B1 [75]. Defects in autophagy due to monoallelic loss of Beclin-1 or downregulation of Atg5 contribute to increased DNA disruption, which may contribute to an oncogenic event in the cells [76]. Ultra-violet (UV) radiation resistance-associated gene protein (UVRAG) has been found to positively control autophagy by associating with Beclin-1 [77]. The impact of UVRAG-induced autophagy on proliferation, anchorage-independent growth, and tumor initiation indicate that wild-type UVRAG prevents proliferation and tumorigenicity, and serves as a tumor suppressor as well [78]. Endophilin B1 (Bif-1), another recognized tumor suppressor, integrates with Beclin-1 via UVRAG to improve Class III PI 3-kinase complex activity, and further loss of Bif-1 suppresses autophagosome development [79]. p53 has been found to be a known tumor suppressor mutated in several cancers. p53 stimulates AMPK, which leads to the initiation of autophagy through genotoxic burden [80]. In addition, p53 triggers autophagy by enhancing DRAM expression [56]. The loss of p53 in the cytosol, however, also leads to autophagy [59]. The p53 has to be further established as both a stimulator and an autophagy antagonist. Atg4C, a protease, acts as an ROS sensor in the core autophagic machinery, which facilitates autophagosome formation via LC3/Atg8 processing [81]. Atg4C knockout mice were more susceptible to fibrosarcoma development [82]. The significantly weaker tumor suppressor role of Atg4C needs further research to establish the key function of the autophagy mechanism in tumor suppression. Nonetheless, in hematological tumorigenesis and solid malignancies, particularly liver cancer, autophagy defects are observed [4]. The function of interferon-γ (IFN-γ) with proliferation inhibition on HCC has been shown by Li et al. [83]. In addition, controlled autophagy induced proliferation inhibition and cell death in Huh7 HCC cells. They also showed that IFN-γ induced autophagosome development and facilitated shifts in autophagic signals, and the autophagic flux contributes to non-apoptotic cell death. Subsequently, blocking autophagy did not show any proliferation inhibition and cell death effects on HCC [83]. This evidence suggests that the role of autophagy as a tumor suppressor is regulated by scavenging or avoiding ROS aggregation in order to maintain genome stability.

### 2.4. Autophagy-Mediated Tumor Promotion

Tumor cells usually have high rates of proliferation that require high autophagy to deal with oxidative stress and deprivation of nutrients [84]. Increased autophagy activity with an invading phenotype and weak prognosis occurs in advanced human tumors. Relevantly, elevated levels of basal autophagy are observed in tumor cells in which oncogene Ras is triggered [85]. Metabolic stress often exists in primary tumors with inadequate blood supply or perhaps in cancer cells, which metastasize to organs with a limited supply of nutrients [86]. Autophagy is capable of providing nutrients through cytoplasm breakdown, which allows the tumor cells to live under pressure and the idea is that autophagy upregulation promotes the survival mechanism of metastasizing cells during cell detachment from the extracellular matrix [87]. Indeed, autophagy was suggested as the response of the tumor cells to survive under chemotherapy- or radiotherapy-induced metabolic stress [88]. Long non-coding RNAs (lncRNA) are involved in carcinogenesis of the liver facilitated by autophagy. lncRNA is generally highly upregulated in HCC, resulting in the upregulation of the genes Atg5 and Atg7, thereby facilitating the proliferation of tumor cells [89]. Another study showed that the upregulation of p62 expression is essential to activate nuclear factor erythroid 2-related factor 2 (Nrf2) and mTORC1, leading to c-Myc activation and to defend the HCC cells against oxidative stress [90]. Further, Beclin-1, Atg5, and Atg7 deletion was found to be correlated with the HCC phenotype [91]. The outcomes of the loss of essential autophagy genes indicate that basal autophagy plays an important role in avoiding the emergence and growth of tumors in the liver. Autophagy facilitates the progression of liver cancer by inhibiting tumor suppressor expression or by leading to chemotherapy resistance in cancer cells. Taken together, during tumor progression, autophagy can favor for survival, malignant growth, and remote metastasis of tumor cells. The induction of autophagy in developed cancer cells can be an effective anticancer mechanism.

### 2.5. Mutated/Dysregulated Autophagy Genes/Proteins in Cancer

Sufficient findings suggest that autophagy can either inhibit or promote cancer, which depends on the tumor type. The autophagy role likely differs in different stages of cancer development as downregulation of autophagy-related genes Atg5 results in reduced growth of metastatic carcinoma, whereas Atg7 promotes apoptosis of colon cancer cells [92]. In mammals, Beclin-1 (encoded by the *BECN1* gene) plays a vital role in autophagy. Monoallelic deletion of the *BECN1* gene has been discovered in human prostate, ovarian, and breast cancers [93,94,95]. Furthermore, Beclin-1’s aberrant expression correlates with poor prognosis for different tumor types, such as HCC [96,97,98]. Beclin-1 interacts with PI3K class III lipid-kinase complex in autophagy, positively regulated by UVRAG [78]. Monoallelic mutated UVRAG in human colon cancers is associated with fostering autophagy and also suppresses human colon cancer cell proliferation and tumorigenicity. These findings suggest that UVRAG is an important indicator of autophagy and the growth of tumor cells [78]. EI24/PIG8 autophagy-associated transmembrane protein has also been known to play a role as pro-apoptotic and tumor suppressor function, which is reported to be mutated in breast cancer cells [99]. In addition to Beclin-1 and EI24, changes in the expression of Atg5 proteins and somatic mutations of the Atg5 gene are observed in gastrointestinal and prostate cancers [100,101]. Furthermore, Atg5 is often decreased in primary melanomas, leading to a decrease in basal autophagy function as verified by a reduced expression of LC3. Downregulation of Atg5 therefore results in tumorigenesis in the early skin melanoma, and expression of Atg5 and LC3 proteins correspond with melanoma diagnosis and prognosis [102] (Table 1).

As discussed earlier, mTOR is a serine/threonine kinase that regulates a wide range of cellular functions, including growth, proliferation, and autophagy. The catalogue of somatic mutations in the cancer (COSMIC) database listed that 10 somatic mutations in mTOR are observed in melanoma, lung, ovary, colorectal, brain, and kidney cancer cells [104]. In addition, well-studied tumor suppressors, such as PTEN (an Akt inhibitor), tuberous sclerosis 1 (TSC1), and TSC2 (mTOR inhibitors), give signals to stimulate autophagy, showing that raised autophagy signaling results in tumor suppression. The aforementioned evidence showed a strong link between the development of cancer and the shifts correlated with autophagy. Remarkably, in many instances, tumor specimens displayed downregulation of an autophagy gene rather than amplification and upregulation. The prevailing view of autophagy as a modern tumor suppression mechanism is further supported by the fact that many anticancer agents function as effective autophagic inducers.

### 2.6. Molecular Mechanisms and Biomarkers of Autophagy

Promising results from several preclinical studies have led to the initiation of numerous clinical trials with autophagy targeting. Autophagy has a role in promoting tumor initiation, development, survival, maintenance, malignancy, and metastasis in various cancers [71]. Hence, autophagy has become a successful target for anticancer therapy. Various autophagy proteins have favorable prognostic value in oncology, for example, LC3B in non-small cell lung cancer (NSCLC) [111] and breast cancer (BC) [112]; LC3A in colorectal cancer (CRC) [113] and BC [114]; p62, and ULK1 in BC [115] and gastric cancer (GC) [116]; Atg3 in GC [116]; Atg10 in GC [116]; Atg7 and FIP200 in BC; and Beclin-1 and VPS34 in GC, BC, NSCLC, CRC, and lymphoma [117]. p62 accumulation was observed in pre-malignant liver disease and high p62 expression indicates poor prognosis in HCC [90]. Mouse models of HCC revealed that p62 is required to activate TSC2-driven mTORC1and Nrf2.

The exact autophagy manipulation by the United States Food and Drug Administration approved anti-HCC drug sorafenib in HCC treatment is not well understood. However, enough evidence showed that sorafenib is able to promote autophagy-mediated cell death by overexpression of autophagy markers, including Beclin-1, by induction of ER stress and by suppression of the autophagy inhibitor mTORC1 and Akt [118]. Another study showed that the analysis and comparison of combined autophagy biomarkers, such as ULK1 and LC3B, with patient prognosis would better represent the complex stage of autophagy [119]. Wu et al. [120] showed that high LC3B expression is a novel prognostic biomarker in HCC patients, which was associated with vascular invasion and lymph node metastasis. The autophagic LC3A gene is expressed differently in human cancers. Examination based on tissue microarray-based immunohistochemistry revealed that high levels of LC3A expression were observed more frequently in HCC tissues compared to the adjacent non-tumor tissue. Furthermore, high LC3A expression was correlated with higher serum α-fetoprotein levels, lower tumor differentiation, and vascular invasion in HCC [121]. The other study investigated the relationship between ULK1 expression and clinical pathology characteristics as well as survival in patients with HCC. ULK1 levels of expression in HCC and adjacent tissues were analyzed using immunohistochemistry, indicating that ULK1 expression in adjacent tissue was lower than that in HCC tissues and high expression is associated inversely with survival. In addition, ULK1 expression is also significantly associated with tumor size [122]. The results above provide evidence that proteins associated with autophagy function as a possible novel biomarker of prognosis for HCC patients.

### 2.7. Autophagy Manipulation in HCC Therapy

Despite significant progress in HCC treatment, many cancers still exhibit poor response and are resistant to therapy. Currently, most of the available HCC therapies besides surgery include chemotherapy and the use of biological (as hormones and antibodies) and physical agents (as radiotherapy) [123]. Basal autophagy functions as a tumor suppressor during the dysplastic process of hepatocytes by removing newly impaired organelles and thereby retaining cell homeostasis. Nevertheless, once a tumor is detected, unbalanced autophagy may lead to the survival of HCC under specific stress conditions, which, in effect, will stimulate tumor growth [124]. The key autophagic genes deliver an important position in preventing the development and occurrence of HCC. It was observed that the deletion of Beclin-1, Atg5, and ATtg7 was correlated with random malignancies like HCC [91]. Meanwhile, differential expression levels of Beclin-1 correlated with the development of HCC. The Atg5-deficient mice hepatocytes derived tumors in the liver alone, which revealed the hepatic-specific antitumor effect of autophagy [125]. The major factors in the oncogenesis of the liver cells are recurrent disruption of the cells and genetic abnormalities. Moreover, loss of autophagy contributes to p62 accumulation, resulting in disturbed genetic stability to form a hepatic tumor [126].

lncRNA PTENP1, a pseudogene of tumor suppressor gene PTEN, is downregulated in various HCCs [127]. Increased levels of PTENP1 and PTEN blocked the oncogenic PI3K/Akt pathway and autophagy, thus inhibiting the survival of HCC [127]. Peiyuan et al. [83] demonstrated that autophagy mediated the proliferation inhibition and cell death in HCC cells as well as caused proliferation inhibition. In contrast, IFN-γ induced autophagosome development and facilitated improvements in autophagic signaling and autophagic-mediated HCC cell death. Sorafenib is a multi-kinase inhibitor that targets multiple genes for the modulation of cancer cell survival, proliferation, and metastasis. However, Shi et al. [118] showed that sorafenib was found to induce apoptosis and autophagy in human HCC cells by upregulating inositol-requiring enzyme (IRE1) signals involving ER stress. Sorafenib also triggers autophagy by activating the Akt pathway or inhibiting mTORC1 [128]. The reduced activation of autophagy can inhibit the HCC, while high autophagy activation contributes to HCC survival [129]. In clinical trials, rapamycin therapy in post-liver transplantation HCC patients showed increased survival rates [130]. Furthermore, sirolimus, a rapamycin analog, showed similar findings with advanced HCC conditions, but without hepatic transplantation, and everolimus exhibits tumor inhibition in HCC xenograft models [131,132]. Some of the potent inhibitors identified to arrest HCC cells were ULK1 small molecule inhibitor (SBI-0206965), the PI3K/mTOR inhibitor (BEZ235), and it was further found that SBI-0206965 and everolimus or BEZ235 synergized with the mTOR inhibitor to kill HCC cells [133]. The autophagy inhibitor hydroxychloroquine (HCQ) exerts tumor suppression, which suppresses autophagy by inhibiting lysosome function when evaluated in mouse models [134]. Interestingly, autophagy inhibition by HCQ in cancer stem-like cells affected the invasion and migration of HCC cells. To date, multiple clinical trials have been carried out to examine whether sorafenib/HCQ controls autophagy and whether this combination will have increased efficacy in HCC care relative to sorafenib alone. Jian et al. [135] demonstrated that a novel oncolytic adenovirus targeting Wnt effectively inhibited cancer stem cell development, such as metastasis, apoptosis, and autophagy, in HCC models. Further, combination chemotherapy based on oxaliplatin has been shown to induce autophagy in HCC patients that correlated with increased ROS stress [136]. Similarly, advanced HCC patients who were treated regularly with doxorubicin to trigger autophagy by promoting the MAPK/ERK pathway contributed to autophagic development preceding cell death [137]. Hence, it may be proposed that the inhibition of various signaling pathways of autophagy directly or indirectly could participate in treating the occurrence and progression of HCC.

Accumulation of studies shown that many cancers are associated with oncogenic mutations, which result in RTK overexpression or suppression [138]. The combination of autophagy antagonists and RTK inhibitors demonstrates an enhanced reaction towards treatment in cell culture and xenograft models. In addition to sorafenib, another tyrosine kinase inhibitor (TKI), nilotinib, triggered autophagy in HCC cells through AMPK phosphorylation [139]. Further HCQ inhibition by autophagy reduced the nilotinib effect in an in vivo model [139]. Further, sorafenib, a multikinase inhibitor, is the current systemic agent for the first-line therapy of advanced HCC. However, the efficacy and therapeutic duration remain limited by an acquired or inherent resistance. Regorafenib, a multiple kinase inhibitor, is to be licensed for the treatment of advanced HCC patients who have recovered after sorafenib therapy. Regorafenib substantially improved progression-free survival, time to development, and also reasonable tolerability [140]. In Hep3B and HepG2 cells, Brian et al. [141] assessed regorafenib- and sorafenib-induced apoptosis and autophagy mediated cell death through the cellular MAPK signaling pathway. Lenvatinib is a TKI and is licensed in the USA, EU, Japan, and China for the first-line treatment of patients with unresectable HCC. Given its manageable tolerability profile, lenvatinib represents an alternative option to sorafenib [142]. Cabozantinib is a TKI used for the treatment of HCC patients who have previously been treated with sorafenib, and is recommended in the EU and USA, as a tablet [143].The first-line and second-line systemic drug for advanced HCC has been used for a decade but the treatment spectrum is expanding. However, in the future, we anticipate our increasing perception of small molecule protein kinase inhibitors and autophagy signaling pathways to pave the way for targeted therapy for HCC.

## 3. Role of Polyphenols in Cancer

Plant polyphenols are classified into two major groups, namely flavonoids and non-flavonoids. Flavonoids are heterogeneous polyphenols abundantly present in nature in many vegetables, fruits, and herbs. More than 2000 flavonoids have been isolated from natural sources [144]. The arrangement of the flavonoid has two aromatic rings, A and B, together with a three-carbon connection, usually, in the form of a heterocyclic loop, C. Increases in ring C substitution patterns result in various subgroups of flavonoids, such as flavonols, flavanones, flavones, flavonols, isoflavonoids, anthocyanidins, proanthocyanidins, and lignins [144,145]. The non-flavonoids, phytochemicals containing an aromatic ring with one or more hydroxyl groups, include hydroxybenzoates (phenolic acids and tannins), hydroxycinnamates, saponins as well as stilbenes and their conjugated derivatives. Flavonoids play a vital role in the prevention and treatment of cancer via various mechanisms, including antioxidant, anti-inflammatory, carcinogen-inactivating, antiproliferative, cell cycle arrest-inducing, apoptotic, anti-angiogenic, autophagy-modulatory effects, and reversal of multidrug resistance or a combination of these mechanisms. Flavonoids are involved in both the prevention and treatment of several cancer types. Epidemiological data strongly suggests that naturally occurring flavonoid compounds greatly reduce the risk of many cancers, such as HCC [146]. Several anticancer agents derived from plants, including paclitaxel, docetaxel, camptothecin, and its derivatives, vinblastine, vincristine, topotecan, irinotecan, and etoposide, are used in cancer treatment [6,7]. Flavonoids and their chemical analogs have shown their potential role in vaginal, breast, cervical, pancreatic, and prostate cancer treatment [21].

While luteolin and kaempferol can be considered as promising candidate agents for treatment for gastric and ovarian cancer, respectively, apigenin, chrysin, and luteolin have good potential as antitumor agents for cervical cancer [21]. The anticancer effect of flavonoids toward hematological cancer cells depend on their myeloid, lymphoid, or erythroid origin; cytotoxic effects of flavonoids on breast and prostate cancer cells are highly related to the expression of hormone receptors [21]. Wang et al. [147] found that rottlerin (a polyphenol) exhibited its anticancer activity via upregulating DEAD-box RNA helicase 3 (DDX3) expression, p21 level, and downregulating cyclin D1 expression in HCC cells. DDX3, which belongs to DEAD-box family proteins, plays a critical role in cancer development and progression, and also DDX3 is a tumor suppressor in HCC [148]. Silymarin from milk thistle seeds, *Silybummarianum* L. Gaertn., contains silibinin, which consists of a mixture of two flavonolignans called silybin A and silybin B. It has various therapeutic effects, such as antioxidant, anticancer, immunomodulatory, antiviral, and antifibrotic, in different tissues and organs [149]. Numerous studies stated that silymarin has anti-HCC potential without affecting the non-tumor hepatic cells [150]. Silymarin reduced the percentage of cells in the S-phase associated with downregulation of cyclin E, cyclin D1, phospho-Rb, and CDK4 and upregulation of p53, p27Kip1, and p21^Cip1^ [151]. Ramakrishnan et al. [150] described that silymarin treatment with HepG2 cells resulted in cell cycle arrest, anti-proliferation, decreased mitochondrial transmembrane potential, and leads to apoptotic cell death, through increased expression of p53, Bax, APAF-1, and caspase-3 (pro-apoptotic) proteins, decreased expression of Bcl-2 (anti-apoptotic), and decreased regulation of β-catenin, cyclin D1, c-Myc, and proliferating cell nuclear antigen (PCNA). Silymarin was also demonstrated to have a dose-responsive preventive role and leads to hepatic tissue regeneration through repairing early stage hepatic damage [152]. Further, the use of silibinin in rats was protective against diethylnitrosamine-induced HCC [153].

## 4. Polyphenols as Modulators of Autophagy in Cancer

Global research focuses on discovering novel natural phytochemicals with autophagy-modulating properties as potential candidates for cancer treatments with minimal side effects. Many synthetic compounds as modulators of autophagy have also been reported as potential candidates for cancer therapy. Natural polyphenolic compounds, such as genistein, quercetin, and rottlerin, can modify the molecular mechanism and trigger cell death through autophagy. Rottlerin could be used to induce autophagic cell death apoptosis in prostate cancer stem cells via the PI3K/Akt/mTOR signaling pathway [154]. Further, rottlerin induces autophagy cell death via the PKC-δ-independent pathway in HT1080 human fibrosarcoma cells [155] and autophagy-mediated apoptosis in breast cancer stem cells [156]. Genistein induces autophagy by modulating the antioxidants proteins that trigger cell death in human breast cancer cells MCF-7 [157]. Quercetin exhibited an anticancer property via stimulating autophagy by interfering with several pathways related to cancer, such as PI3K/Akt, Wnt/β-catenin, and STAT3 [158]. Further, quercetin induced autophagy flux, causing lung cancer cell death through the TRAIL signaling pathway [159]. One of the flavonoids, chrysine, blocked temozolomide-induced autophagy and O6-methylguanine-DNA methyltransferase expression in GBM8901 cells and was found to be a potential candidate for glioblastoma cancer [160]. Crysine also induced autophagy by increasing the levels of LC3-II to improve apoptosis in MCF-7 cells [161]. Safe chemotherapy could be an effective therapy for any type of cancer and subsequent metastasis. On the other hand, the occurrence of drug resistance minimizes the treatment effect of chemotherapy. Since autophagy is involved in cancer progression, the manipulation of autophagy using natural compounds could be a promising candidate for multidrug resistance in cancer cells. Nobiletin, a polymethoxy flavonoid, has suppressed human SKOV3/TAX cell proliferation via inducing apoptosis and autophagic flux inhibition. In contrast, through triggering Akt signaling, the disrupted autophagic flux stimulated nobiletin-induced apoptosis in SKOV3/TAX cells. These findings supported nobiletin’s ability to overcome multidrug resistance in human ovarian cancer cells by suppressing autophagic degradation by Akt regulation [162]. The most important for the treatment of advanced NSCLC were EGFR and TKIs. However, T790M mutation, which raised resistance in TKIs caused by EGFR, has become a major challenge in cancer treatment. Zhang et al. [163] revealed that the combined treatment of wogonin and icotinib could overcome T790M-mediated resistance to icotinib. They found an increased number of intracellular autophagosomes, the transformation of LC3B-I to LC3B-II, Beclin 1, and the upregulation of the rates of phosphorylated mTOR expression with the combination of wogonin and icotinib, which suggests that the combination has inhibitory synergistic effects on cell proliferation and could contribute to apoptosis and autophagy in EGFR T790M-mutated lung cancer [163].

Polyphenols also regulate autophagy to overcome or restore multi-drug-resistant cells. For example, apigenin significantly increases the sensitivity of doxorubicin-resistant BEL-7402/ADM cells, stimulates miR-520b expression, and suppresses the AtgG7 [164]. In ovarian cancer cells treated with cisplatin, autophagy regulation leads to the upregulation of PARP-1, which is important for cell survival. Luteolin can inhibit PARP-1 at both mRNA and protein levels, and suppress autophagy, restoring cisplatin tolerance [165]. Scutellarin, an active flavone isolated from *Erigeron breviscapus* (vant.) Hand.-Mazz., decreased the expression of cell cycle-related proteins Cdc2, cyclin B1, and induced apoptosis in PC3 cells, and retained cisplatin resistance [165]. Flavonolignans are natural polyphenolic compounds, which consist of a flavonoid and a lignin part. Silibinin, a flavonolignan derived from silymarin [*Silybummarianum*(L.) Gaertn] extract, exhibited antigenotoxic activities, membrane-stabilizing and antioxidant activity, stimulated hepatocyte regeneration, inhibited the fibrogenesis, and reduced the inflammatory response in the liver. Long-term intake of silymarin suggestively improved the survival rate of patients with liver cirrhosis [166,167]. Silibinin is reported to induce autophagy in HeLa cervical carcinoma [168], and MCF-7 breast cancer cells by the formation of LC3-II, Atg12-Atg5, and also upregulating Beclin-1 [169]. Further, silibinin induced autophagy in human fibrosarcoma HT1080 cells by activation of the p53 by ROS/p38/JNK pathway [170] in melanoma A375-S2 cells [171] and human colon cancer cells SW480 and SW620 [172]. However, silibinin exerted its anticancer effect via the apoptotic pathway in HCC cells, such as Huh7, HepG2, Huh-BAT, Hep3B, HepG2, PLC/PRF5, SNU387, SNU398, SNU449, SNU475, and SNU761cells, and also exhibited growth-inhibitory activity with either sorafenib or gefitinib via a synergistic manner in the SNU761 cell line [173]. Since silibinin is an effective inducer of autophagy in various cell lines, its role as an autophagy-mediated anti-HCC effect needs to be explored further.

Nonetheless, the absence of insufficient testing of animal models, treatment protocols, and phytochemical toxicity analyses undermines the inherent accuracy of the individual studies, making it difficult to determine the efficacy of plant-derived compounds in the treatment of certain cancer types.

## 5. Polyphenols Targeting Autophagy in HCC

Autophagy is a tightly regulated conserved process that involves the degradation of intracellular unwanted components via fusion with lysosomes. The critical role of autophagy in cancer is still controversial, however, both tumor-suppressing and tumor-promoting functions have been reported. Several studies have shown that flavonoids and non-flavonoid polyphenols can induce autophagy, both in vitro and in vivo, and exhibited their anti-HCC effect. Flavonoids, such as quercetin, apigenin, and epigallocatechin gallate (EGCG), demonstrated antiproliferative activity by upregulating the expression of Beclin-1, LC3-II, and several forms of Atg in several HCC cells. Cheng et al. [174] found that the extract of mulberry fruit, containing rich polyphenols, could inhibit diethylnitrosamine-induced hepatocarcinogenesis. The extract of mulberry polyphenol mediated autophagy in cells of Hep3B by inhibiting phosphorylation of Akt and mTOR [174]. However, a more specific understanding is needed for HCC therapy that addresses the modulation of autophagy utilizing polyphenols.

### 5.1. Flavonoids

#### 5.1.1. Flavones

Apigenin (Figure 2), one of the most studied cancer preventive and therapeutic agents, regulates a variety of intracellular signal transduction pathways associated with autophagy or apoptosis in HCC [175]. In HepG2 cells, apigenin triggered apoptosis and autophagy via the PI3K-Akt-mTOR signaling pathway and by decreasing the level of the autophagy substrate sequestosome-1 (SQSTM1) [176] (Table 2). Apigenin also increased the expression of LC3-II and mediated protective autophagy. The effect of apigenin to prevent HCC tumor growth was also proved in the presence of the autophagy inhibitor 3-methyladenine (3-MA) [176]. Oroxylin A (an O-methylated flavone) caused autophagic cell death through Beclin-1 expression in human HCC HepG2 cells, as well as the formation of water-soluble LC3-I after 12 h to LC3-II, then autophagosome-lysosome fusion and lysosome degradation after 24 h. This activation was consistent with oroxylin A-mediated inhibition of the PI3K-PTEN-Akt-mTOR signaling pathway [177].

The hepatoprotective effect of isorhamnetin was evaluated in mice by inhibition of apoptosis and autophagy through the p38/PPAR-α pathway [178]. However, isorhamnetin acted against liver fibrosis by increasing the development of ECM and autophagy by inhibiting TGF-β1-mediated Smad3 and p38 MAPK signals [179]. Baicalin has been shown to cause autophagic cell death in the SMMC-7721 cells with CD147 downregulation [180]. Another study showed that baicalein extracted from the medicinal herb *Scutellariae radix* exhibited mild toxicity on HepG2 cells. While baicalin and chloroquine combined treatment decreased cell viability and colony formation, baicalin triggered protective autophagy that prevented cell death [181]. Tangeretin (5,6,7,8,4’-pentamethoxyflavon) decreased cell proliferation and increased G2/M arrest by increasing the LC3II/LC3I ratio and decreasing p62 in HepG2 cells [182]. In addition, the Beclin-1 expression knockdown induced partial proliferation and autophagy inhibition. Furthermore, tangeretin triggered the JNK1/Bcl-2 pathway and disturbed the Bcl-2-Beclin-1 interaction. It was shown that tangeretin blocked JNK/Bcl-2/Beclin-1 proliferation [182]. Sorafenib, a multi-kinase inhibitor, is used as an oral agent for the treatment of advanced HCC and renal cell carcinoma. However, earlier studies revealed that it may contribute to cancer cell progression and lead to sorafenib resistance by activating autophagy. Therefore, inhibiting autophagy via combination therapy could improve sorafenib-induced cell death. Rong et al. [183] reported that in human HCC cell lines Hep3B, Bel-7402, HepG2, and SMMC-7721, the combination of sorafenib and wogonin exerted substantial cytotoxicity, in which wogonin inhibited autophagy triggered by sorafenib, which potentiated apoptosis in HCC.

Isoorientin or homoorientin is a flavone C-glycoside reported to induce autophagy by the formation of autophagy vacuoles and expression of Beclin-1 and LC3-II in human HepG2 cells [184]. The autophagy inhibitor 3-MA markedly inhibited apoptosis, and the apoptosis inhibitor ZVAD-fmk also decreased isoorientin-induced autophagy. Moreover, the Beclin-1, LC3-II, and PARP cleavage levels were increased by the PI3K/Akt inhibitor LY294002. Likewise, N-acetyl-L-cysteine (a ROS blocker), SP600125 (JNK inhibitor), and SB203580 (p38 inhibitor) effectively reduced the concentrations of these proteins. Furthermore, the p53 inhibitor pifithrin-α and the NF-κB inhibitor pyrrolidine dithiocarbamic acid suppressed autophagy-associated proteins, including Beclin-1 and LC3-II, and increased cytochrome c release, caspase-3 activation, and PARP cleavage. These findings confirmed that isoorientin synchronously induced autophagy and apoptosis by way of ROS-related p53, JNK, PI3K/Akt, and p38 signaling pathways in HepG2 cells [184]. Luteolin mediated apoptosis in a time and concentration-dependent fashion in HCC cells (SMMC-7721), increased the number of intracellular autophagosomes, facilitated the transformation of LC3B-I to LC3B-II, and increased the expression of Beclin-1. Co-treatment with the chloroquine (an autophagy blocker) also reduced luteolin’s effects on cell apoptosis [185].

#### 5.1.2. Flavonols

Galangin, obtained from medicinal herb *Alpiniaofficinarum* (Hance), induced autophagy through a p53-dependent pathway in HepG2 cells. In general, galangin therapy in HepG2 cells induced aggregation of autophagosomes, high levels of the protein LC3 associated with microtubules, and increased the number of cells with vacuoles. In addition, galangin-induced autophagy was attenuated by p53 inhibition in HepG2 cells and overexpression of p53 in Hep3B cells restored the higher levels of galangin-induced cells with vacuoles to normal levels [186]. Wang et al. [187] showed that galangin induced autophagy through triggering TGF-βRI and TGF-βRII, in addition to activating Smad1, Smad2, Smad3, and Smad4 levels, but decreased Smad7 and Smad6 levels. Further decreased levels of Beclin-1 expression and increased expression of autophagy-related genes, such as Atg3, Atg16L, and Atg12, prevented galangin-induced apoptosis on HCC cells [187]. Kaempferol was also documented to impede tumor effects in SK-HEP-1(HCC). Kaempferol did not cause DNA fragmentation, caspase-3 activity in SK-HEP-1 cells, but through transmission electron microscopy, kaempferol was found to be involved in the autophagic cycle. The study revealed double-membrane vacuoles, lysosomal compartments, acidic vesicular organs, and cleavage of protein LC3 along with microtubules. In contrast, kaempferol-treated SK-HEP-1 cells showed elevated protein levels of p-AMPK, LC3-II, Atg 5, Atg 7, Atg 12, and Beclin-1 and suppressed protein levels of cyclin-dependent kinase 1 (CDK1), cyclin B, p-Akt, and p-mTOR. Taken together, the expression of CDK1/cyclin B and the signaling pathways for AMPK and Akt led to the arrest of kaempferol-induced G2/M cell cycle and autophagic cell death in SK-HEP-1 cells [188]. In HepG2 and Huh7 cells, Guo et al. [189] established that kaempferol caused autophagy. Kaempferol-induced apoptosis was severely affected during treatment with the apoptosis inhibitor 3-MA or Atg7 siRNA. siRNA-mediated knockdown of CCAAT/enhancer-binding protein homologous protein (CHOP) reduced kaempferol-induced HepG2 or Huh7 cells’ autophagy, which revealed that kaempferol triggered HCC death through the CHOP autophagy signaling pathway [189].

Recent literature on quercetin (3,3′,4′,5,7-pentahydroflavone) has elucidated that its cytotoxic effect is associated with apoptosis and autophagy induction. Liwei et al. [190] showed that quercetin exhibited significant antiproliferation effects on HCC LM3 cells and xenograft mice tumor models. They observed that quercetin controlled LM3 cells by increasing LC3 expression and downregulating P62 function [190]. Furthermore, Ji et al. [191] revealed from the results of electron microscopy, confocal fluorescence GFP-RFP-LC3, and Western blot analysis that quercetin increased the formation of autophagosomes and autolysosomes in both in vitro and in vivo HCC models. In addition, autophagy stimulation was supported by function-specific inhibitors or siRNAs in HCC cells [191]. The chemopreventive flavonoids, such as quercetin, kaempferol, and galangin, induced ROS signaling, cell cycle arrest, and promoted cancer cell apoptosis via the induction of the initial autophagic process, which has been reported also for myricetin in liver cancer. Myricetin induced autophagy by inhibiting the phosphorylation of mTOR activation in HepG2 cells [192].

#### 5.1.3. Flavanols

Flavanols represent the most complex subclass of flavonoids. The major flavanol compounds present in green tea are catechins, such as EGCG, which has been investigated for its anti-cancer ability in HepG2 cells, in which autophagosome formation was recorded by transmission electron microscopy [193]. HepG2 cells treated with EGCG also significantly reduced the transcript and protein levels of Beclin-1 and Atg5, with increased levels of autophagic substrate P62 [193]. In addition, Chen et al. [194] demonstrated the synergistic effects of EGCG and doxorubicin on HCC therapy concerning autophagic flux in general. Combined with EGCG, doxorubicin played a concentration-dependent inhibitory role in autophagy signaling through increased Beclin-1 and Atg5 expression to inhibit autophagy and suppressed LC3 expression in hepatoma Hep3B cells [194].

#### 5.1.4. Anthocyanidins

Anthocyanins are part of the phytochemical flavonoid group, which comprises, cyanidin, delphinidin, pelargonidin, and petunidin. They are the natural pigments and the different substituent classes vary depending on the flavylium B-ring. The capacity of anthocyanins to suppress tumor development and cancer cells has been shown in clinical trials. The anthocyanin pigments have the potential to interfere with the process of carcinogenesis, which appears to be linked to multiple mechanisms of action and potent antioxidant property. Feng et al. [197] reported that delphinidin was able to induce apoptosis and antiproliferation in SMMC7721 cells by promoting cell vacuolization. Delphinidin also caused extensive lipidation of LC3 II, an autophagy signal required for the formation of autophagosome [197].

### 5.2. Non-Flavonoids 

#### 5.2.1. Stilbenes

Resveratrol, a natural polyphenol, did not show autophagic response at a low concentrations (10 µg/mL); however, at a higher concentration (20 µg/mL), it activated autophagic cell death through increased autophagy-related expression of Atg5, Atg7, Atg9, and Atg12 proteins in Huh 7 cells [198]. Zhang et al. [199] reported that resveratrol exerted antitumor effects in the HCC MHCC97-H cell line in a time- and concentration-dependent manner. Moreover, resveratrol treatment elevated the autophagy-related protein Beclin-1 and LC3 II/I ratio, whereas it decreased p62 expression. Further, treatment with inhibitors, such as 3-MA and pifithrin-α, and insulin-like growth factor-1 (IGF-1) decreased the Beclin-1 expression, with the promotion of MHCC-97H cell survival. Altogether, resveratrol exhibited an anti-HCC effect by inducing autophagy through inhibiting PI3K/Akt and activating p53 [199].

#### 5.2.2. Hydroxycinnamates

E-[6′-(5′-hydroxypentyl)tricosyl]-4-hydroxy-3-methoxycinnamate (EHHM)), a phenolic natural product obtained from *Livistona chinensis*, revealed inhibition of cell proliferation and colony formation in HepG2 cells. Meanwhile, in HCC cells, EHHM triggered the expression of the proteins, such as Atg5, Beclin-1, and LC3-II. Additionally, treatment with siRNA for Atg5 or autophagy inhibitors promoted cell inhibition. It also suggests that autophagy improves cell survival in HCC cells treated with EHHM and that the use of EHHM could be an effective approach in the treatment of HCC [200].

#### 5.2.3. Miscellaneous Non-Flavonoids

A plethora of publications have shown that curcumin, a phytopolyphenol, is the most extensively studied natural compound for cancer prevention and treatment [206]. Curcumin-treated Huh7 cells induced early autophagy, characterized by the development of autophagic vacuoles due to the conversion of LC3-I into LC3-II [201]. Curcumin also increased the survival rate against thioacetamide-induced HCC through activating the autophagy signaling cascade by the expression of proteins, such as LC3-II and SQSTM1, and inhibiting apoptosis in male Sprague-Dawley rats [202]. The combination of curcumin therapy with adriamycin (doxorubicin) caused autophagy activation with elevated protein expression of LC3-II up to 48 h after drug administration against HepG2 cells. In the juxtanuclear region, high concentrations of the autophagosomes were observed in the cells treated with this synergistic combination. Studies carried out with 3-MA, the autophagy inhibitor, also proved the autophagy-inducing role of the combined treatment [203].

#### 5.2.4. Analogs of Non-Flavonoids 

Curcumin has antitumor effects in several cancer types, including HCC. However, curcumin has limited therapeutic potential due to its poor efficacy and bioavailability. Hence, various analogs of curcumin have been synthesized to evaluate its anticancer potency. The cytotoxicity activity of water-soluble curcumin analog, 3,5-bis(2-hydroxybenzylidene)tetrahydro-4H-pyran-4-one glutathione conjugate [EF25-(GSH)_2_] in HCC was assessed by Zhou et al. [204]. EF25-(GSH)_2_ was demonstrated to have limited cytotoxicity against HL-7702 cells and exhibited suppressed growth of HepG2 xenografts in mice. EF25-(GSH)_2_-induced autophagy and further when combined with chloroquine autophagy blocker, significantly increased cytotoxicity in HCC [204]. A novel curcumin derivative, 1-(4-hydroxy-3-methoxyphenyl)-5-(2-nitrophenyl)penta-1,4-dien-3-one (WZ35), was found to mediate the ROS/JNK-dependent signaling cascade and ER stress and act as a potent anti-gastric cancer agent. Wang et al. [205] tested the in vitro and in vivo antiproliferative role of WZ35 in HCCLM3 cells. The authors observed the opposite effects of curcumin and its derivative. For instance, during curcumin treatment, upregulation of LC3 I/II was noticed in HCCLM3 cells, whereas a reduction in the levels was observed during WZ35 treatment. A similar effect was observed for the other autophagic proteins, including Atg 67, p62, and Beclin-1, which substantiates that the derivative might have a better autophagy regulatory activity than curcumin and could contribute to better activity against the HCC cells. The derivative inhibits autophagy mainly through decreasing the expression of the autophagy modulator Yes-associated protein (YAP) [205].

## 6. Clinical Trials

Impairment in autophagy is the major cause of many illnesses from cancer to neurodegenerative disorders. Consequently, the number of studies centered on autophagy regulation as a perspective and potential therapeutic target is steadily rising. Operational resection or liver transplantation has been shown to be successful in the early stage of HCC. However, HCC patients reportedly had decreased survival time following surgery with poor prognosis, recurrence, and metastasis. Effective therapy is currently not available for patients at the late stage of HCC, because of inadequate chemotherapeutic agents, leading to low overall survival rates. In maintaining liver homeostasis, autophagy plays multiple roles. Sorafenib is the only molecular-targeted treatment approved by the FDA as a first-line for early or advanced HCC, offering extended survival for less than 3 months in a phase III trial conducted in Asia-Pacific region patients [207]. Sorafenib is the antagonist of the molecular signaling pathway RAF/MEK/ERK involved in inhibiting pro-angiogenic factors, such as vascular endothelial growth factor, fibroblast growth factor-2, and receptors of cancer growth. Apatinib, a small molecule kinase inhibitor, is considered as a better alternative for patients who were resistant to sorafenib treatment. A study conducted with a limited number of subjects (22 patients with HCC) showed that apatinib decreased the levels of the HCC biomarker α-fetoprotein, proving its effectiveness in the treatment of advanced HCC [208] However, the later phase III clinical trial, EVOLVE-1, conducted in patients with advanced HCC after sorafenib treatment or in patients with sorafenib intolerance, did not reveal a substantial difference in the overall survival rate when everolimus was given alone [209]. So far, as per the ClinicalTrials.gov website maintained by the National Library of Medicine at the National Institutes of Health, 334 findings have been identified on sorafenib and its use to treat early and advanced stages of HCC. These current clinical trials include autophagy-targeting therapy of HCC patients that could lead to successful care in elderly HCC patients. The major issue with sorafenib-induced autophagy in patients with HCC is possible drug resistance [210]. In order to clarify the role of autophagy-induced therapy in the treatment of HCC, more work is therefore needed. In addition, autophagy inhibition could also be a possible therapeutic approach for HCC therapy, as autophagy is important for HCC cell survival, particularly in the early stages. Despite significant advances in the diagnosis and treatment of HCC over the past few decades, late-stage HCC care, recurrence, and insufficient medical approaches to avoid metastasis are still less successful [211]. Regarding polyphenols, though many compounds are reported to regulate autophagy, in order to fully extend the therapeutic properties of polyphenols in HCC therapy, adequate clinical experiments are needed regarding their function in autophagy manipulation.

## 7. Conclusions

The present review focuses on HCC, the third deadly cancer with a poor prognosis due to inefficiency in the treatment. Researchers believe that combining autophagy modulation with molecular targeted therapy is a successful therapeutic strategy for HCC treatment [212]. Accumulating research over the past two decades has highlighted the importance of autophagy in different human diseases. It is generally believed that modulating autophagy behavior can have an effect on disease processes by targeting specific regulatory players in the central autophagy machinery. Both increased and decreased autophagic signals are observed in tumors, indicating their combined tumor suppressor and oncogenic properties during malignant transformation. In order to develop new therapeutic agents, identifying the main autophagy targets is important. The unique ability of polyphenols to suppress cell proliferation and cause apoptosis or autophagy in HCC has generated a strong interest in their potential for targeted therapy. In summary, polyphenols, including apigenin, oroxylin A, and resveratrol, might inhibit the signaling mechanism for PI3K/Akt/mTOR. Luteolin, isoorientin, quercetin, kaempferols, curcumin, adriamycin (doxorubicin) with curcumin, EGCG, EHHM, delphinidin, EF25-(GSH)_2_, oroxylin A, resveratrol, and kaempferols were involved in LC3 II development, thereby triggering autophagy. Myricetin might inhibit mTOR signaling pathway phosphorylation. Galangin can induce autophagy through the signaling pathway of the TGF-β receptor/Smad. Baicalein may induce protective autophagy associated with negative regulation of CD147 and ER stress. Through inhibiting the autophagy process, wogonin together with sorafenib, WZ35, and tangeretin demonstrate their anti-HCC (Figure 3). Natural dietary polyphenols have numerous bioactivities, including anticancer and autophagy-sensitizing effects. Studies show that polyphenolics exert their anti-HCC capability through autophagy manipulation, including stimulation of Beclin-1, Atg5, Atg7, Atg9, Atg12, LC3-II, and SQSTM1, as well as manipulation of PI3K/Akt/mTOR, PTEN, P38/PPAR-α, JNK/Bcl-2, ER stress, p62, p53-dependent, TGF-β receptor/Smad signaling, and YAP, to exert their sensitizing effects, implying that they are potential multi-functional agents for treatment of HCC, which is a devastating disease. One important consideration with the FDA-approved HCC drugs like sorafenib is the safety and potency profiles of these drugs, due to which patients experience serious adverse side effects, including hepatotoxicity, inflammation, hemorrhage, fistula, dermatological risk, hypertension, infarction, and risks of wound healing. Likewise, reports on the efficacy of the available HCC drugs in patients with more severe cirrhosis are not available. Hence, it will be a better alternative to pair autophagy-modulating hepatoprotective polyphenolic compounds (with good safety profiles) with FDA-approved anti HCC drugs that can provide novel therapeutic strategies in the treatment of HCC by the manipulation of autophagy. In addition, the advancement of novel therapeutic strategies must take autophagy-modulating polyphenols in HCC therapy for further investigation.

## Figures and Tables

**Figure 1 cancers-12-00562-f001:**
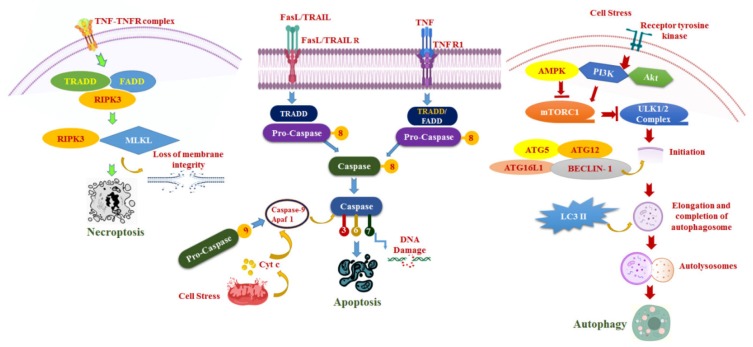
The major cell death pathways: Apoptosis, autophagy, and necroptosis. Apoptosis: DNA injury, stress, toxic accumulation, and nutrient or growth factor deficiency may result in the disruption of mitochondrial membrane integrity to induce the release of cytochrome c from the mitochondria intermembrane space, where a number of Bcl-2 family proteins, such as Bcl-2, Bcl-2-like protein 4 (Bax), Bcl-2 homologous antagonist/killer (Bak), and Bcl-xL, regulate cytochrome c release. Further, cytochrome c can associate with Apaf-1 and caspase-9 to activate caspase-3 and induce apoptosis in the intrinsic pathway. The binding of FasL or TNF-α or TRAIL to their respective receptors will convert procaspase-8 by autohydrolysis into caspase-8. The triggered caspase-8 activates caspase-3, followed by apoptosis in the extrinsic pathway. Autophagy: This process is achieved by induction, development of phagophore and lysosome fusion for degradation. Receptor tyrosine kinases (RTKs) stimulate the PI3K/Akt, which regulates mammalian target of rapamycin complex 1 (mTORC1), which is negatively regulated by 5’-AMP-activated protein kinase (AMPK). The activated AMPK and inhibited mTORC1 leads to activation of unc-51-like kinase 1/2 (ULK1/2) complex, essential for the initiation of autophagy. Beclin-1 complex initiates membrane isolation accompanied by two ubiquitin-like protein-conjugated systems, such as autophagy-related protein 12 (Atg12) and light chain 3 (LC3) systems, involved in autophagosome formation, culminating in lysosome fusion with autophagosome to hydrolyze all unnecessary cellular substances. Necroptosis: The TNF tumor necrosis factor receptor (TNFR) signaling system stimulates the complex consisting of TNFR1-associated death domain protein, FAS-associated death domain protein, receptor-interacting serine/threonine-protein kinase 1 (RIPK1) involved in high-level RIPK3 transition, and mixed lineage kinase-like protein development, resulting in necrosome formation and disruption to the membrane and necroptosis.

**Figure 2 cancers-12-00562-f002:**
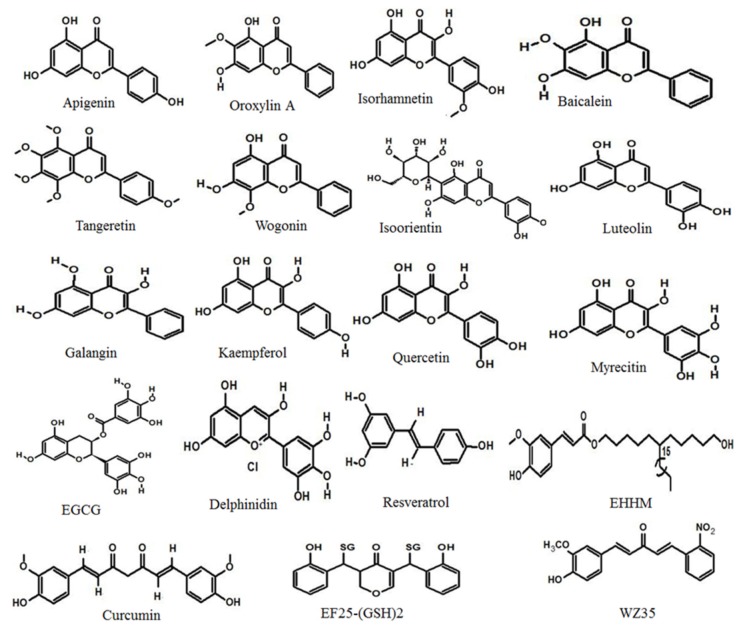
The chemical structures of autophagy-inducing anti-HCC polyphenols.

**Figure 3 cancers-12-00562-f003:**
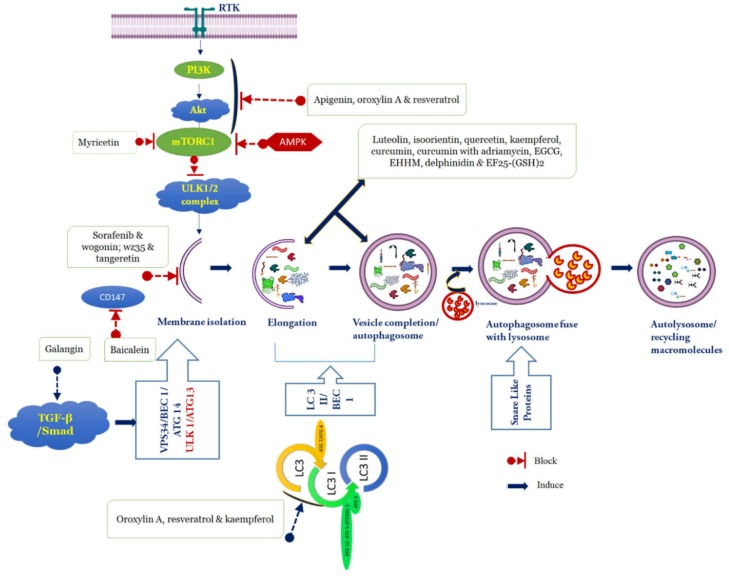
Schematic overview of the autophagy-induced anti-HCC polyphenols signaling pathways. Autophagy-associated signaling mechanism mediated by flavonoids against liver cancer. Apigenin, oroxylin A, and resveratrol could inhibit PI3K/Akt/mTOR signaling pathway. Luteolin, isoorientin, quercetin, kaempferols, curcumin, curcumin with adriamycin, EGCG, EHHM, delphinidin, EF25-(GSH)_2_ involved in the conversion of LC3 II could induce autophagy. Myricetin could inhibit the phosphorylation of mTOR signaling pathway. Oroxylin A, resveratrol and kaempferols might increase the autophagy-related proteins to increase the lipidation of LC3 II. Wogonin along with sorafenib, WZ35, and tangeretin could inhibit the autophagy pathway. Galangin might induce autophagy by TGF-β receptor/Smad signaling pathway. Baicalein may associate with negative regulation of CD147 and mediate protective autophagy through ER stress. Collectively, various polyphenols may treat HCC through the manipulation of autophagy.

**Table 1 cancers-12-00562-t001:** Dysregulated autophagy genes/proteins in cancer.

Genes/Proteins	Function in Autophagy	Alterations in Cancer	Reference
*BECN1*	Autophagosome formation	Monoallelic deletion of the Beclin-1-encoding gene in the human breast, ovarian, prostate, colorectal cancers, leukemia, lung, liver, endometrial, colorectal, glioblastoma and brain cancers	[93,94,95,103]
*EI24/PIG8*	Autophagosome formation and/or degradation	Mutations and deletions are associated with human early-onset breast cancers	[100]
mTOR	Autophagy regulation	Somatic mutation of mTOR in melanoma, lung (large cell), ovary (mucinous), colorectal, brain and kidney cancer cells	[104]
*Atg5*	Autophagic vesicle formation	Alterations of Atg5 protein expression and somatic mutations of the Atg5 gene are found in prostate cancers	[100,101]
*Atg2B, Atg9B, Atg5 and Atg12*	Implementation of the autophagy process; autophagosome formation in early stages autophagy	A frameshift mutation in gastric and colorectal cancers	[105]
*UVRAG*	Nucleation and fusion	Deletion mutation is associated with human colorectal cancer Mutated in gastric and colorectal cancers with microsatellite instability	[105,106]
*p53*	Autophagy regulation	Somatic mutation in human cancer	[107]
*DRAM1*	Autophagy regulation	Substitution mutation in human non-small cell lung carcinoma cell	[108]
*LAMP2*	Mediation of transport of a specific set of cytosolic proteins across the lysosomal membrane in chaperone-mediated autophagy	A missense mutation in pancreatic cancer	[109]
Parkin (E3 ubiquitin ligase)	Autophagy regulation via Bcl-2	A point mutation in ovarian, breast, bladder and lung cancer	[110]

**Table 2 cancers-12-00562-t002:** Autophagy-associated anti-HCC polyphenols.

Polyphenols	Compounds	Cancer Model (Cell line/Animal)	Role in Autophagy	Reference
Flavonoids				
Flavones	Apigenin	HepG2	Inhibited PI3K/Akt/mTOR pathway and downregulated SQSTM1	[176]
	Oroxylin A	HepG2	Caused overexpression of Atg5 and Atg7, inhibition of autophagy by si-Beclin-1 and 3-methyladenine and suppression of PI3K-PTEN-Akt-mTOR signaling pathway	[177]
	Isorhamnetin	Mice	Interfered with p38/PPAR-α pathway	[178]
	Baicalein	SMMC-7721	Displayed downregulation of CD147 and protective autophagy; Protective autophagy via ER stress	[180] [181] [195]
	Tangeretin	HepG2	Interfered with JNK/Bcl-2/Beclin-1 -mediated pathway	[182]
	Wogonin and sorafenib combination	Hep3B & Bel-7402, HepG2 & SMMC-7721	Induced autophagy inhibition	[183]
	Isoorientin	HepG2	Induced overexpression of Beclin-1 and LC3-II, ROS-related p53, PI3K/Akt, JNK, and p38	[184]
	Luteolin	SMMC-7721	Increased Beclin-1 expression and LC3B-II conversion	[185]
Flavonols	Galangin	HepG2 Hep3B	Interfered with p53-dependent pathway and TGF-β receptor/Smad signaling pathway	[186] [187]
	Kaempferol	SK-HEP-1 HepG2 & Huh 7	Increased protein expression of p-AMPK, LC3-II, Atg 5, Atg 7, Atg 12 and Beclin- 1 CHOP-autophagy mediated	[188] [189]
	Quercetin	LM3 Mice	Increased expression of LC3, and downregulated expression of p62; Activated autophagy, Increased the formation of autophagosomes and autolysosomes	[190] [191]
	Myricetin		Inhibited the phosphorylation of mTOR	[192]
Flavanols	EGCG	HepG2	Increased expression of Beclin1, Atg5, increased level of p62 autophagic substrate, promoted the synthesis of LC3-II	[193] [196]
	EGCG and doxorubicin combination	Hep3B	Increased Beclin-1 and Atg5 expression and suppressed LC3 expression	[194]
Anthocyanidins	Delphinidin	SMMC-7721	Exhibited overexpression of LC3-II	[197]
Non-flavonoids				
Stilbenes	Resveratrol	Huh 7 MHCC-97H	Increased the expression of autophagy-related proteins Atg5, Atg7, Atg9, and Atg12 Activated p53 and inhibited PI3K/Akt	[198] [199]
Hydroxycinnamates	EHHM	HepG2	Increased the expression of Atg5, Beclin-1 and LC3-II proteins	[200]
Miscellaneous non-flavonoids	Curcumin	Huh 7 HepG2	Increased the formation of autophagic vacuoles due to the conversion of LC3-I to LC3-II; Induced autophagy with decreased expression of SQSTM1	[201] [202]
	Curcumin with adriamycin (doxorubicin)	HepG2	Increased the expression of LC3-II protein	[203]
Analogs of non-flavonoids	EF25-(GSH)_2_	HL-7702	Induced autophagy	[204]
	WZ35	HCCLM3	Downregulated YAP-mediated and autophagy inhibition	[205]

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
