# Peer review of "Autophagy: A Potential Therapeutic Target of Polyphenols in Hepatocellular Carcinoma"

_cancers, 2020, doi:10.3390/cancers12030562_

Round 1

Reviewer 1 Report

In this article, the authors review the role of autophagy in cancer in general and more specifically in HCC. Also, they review pharmacological strategies to modulate autophagy and its anti-tumor effect, focusing on the possible usefulness of the large and complex group of naturally occurring polyphenols. Although the review is well understood and well written, it is too long in detailing widely known concepts. Moreover, in my opinion, the available experimental data do not justify proposing polyphenols as anti-HCC drugs. The usefulness of drugs that modulate autophagy could be extended to other drugs that have a clinical utility such as tyrosine kinase inhibitors (TKIs), rather than polyphenols.

Major concerns:

1) In my opinion, the review should not focus on polyphenols, because although there are numerous in vitro studies that demonstrate their anti-tumor usefulness, there are hardly any in vivo studies or clinical studies that confirm this. However, there are drugs with potential or real usefulness against HCC, as demonstrated by in vivo studies and clinical trials, which can modulate autophagy, some of which appear in this review (section 6). TKIs should be the drugs on which the review should focus. In contrast, the polyphenol part can be included, much more summarized, but without being the focus of the drugs.

2) In this sense, concerning polyphenols, a multitude of beneficial effects have been described as anti-tumor drugs. What is the importance of their modulating action on autophagy compared to their action on other described pathways?

(3) Paragraph 2, and in particular the subsections on apoptosis, necroptosis and autophagy-regulating signaling pathways, are very extensive and are well-known concepts. They talk about cancer in general. Since the review focuses on HCC, the explanation in these sections should be oriented to HCC.

4) There are currently more TKI drugs that are approved against HCC, such as regorafenib, cabozantinib, and lenvatinib. Only sorafenib and its anti-tumor effects are mentioned in the review. Do these new TKIs effect on autophagy?

Minor concerns:

1) Line 65. “Several anticancer drugs, such as vinblastine, teniposide, camptothecin, docetaxel, etoposide, and paclitaxel, have been identified from natural sources”.

Some of these drugs do not appear in natural sources but are semi-synthetic, obtained by modification of others that do have a natural origin.

2) Some typographical errors in the manuscript need to be corrected.

3) Doxorubicin is the preferred name over Adriamycin.

4) Vinblastin is vinblastine (line 65).

Author Response

The authors of this manuscript express their sincere thanks to the reviewer for the critical assessment of our work. The authors have acted upon the recommendations of the reviewer which have resulted in a significant enhancement of the quality of this manuscript. All modifications incorporated in the manuscript are highlighted using red color font. A “point-by-point” response to the reviewers’ comments is outlined below.

General comments:

In this article, the authors review the role of autophagy in cancer in general and more specifically in HCC. Also, they review pharmacological strategies to modulate autophagy and its anti-tumor effect, focusing on the possible usefulness of the large and complex group of naturally occurring polyphenols. Although the review is well understood and well written, it is too long in detailing widely known concepts. Moreover, in my opinion, the available experimental data do not justify proposing polyphenols as anti-HCC drugs. The usefulness of drugs that modulate autophagy could be extended to other drugs that have a clinical utility such as tyrosine kinase inhibitors (TKIs), rather than polyphenols.

Response:

We thank the reviewer for the valuable feedback. As per the suggestion given, the contents related to the widely known concepts have been reduced. We have excluded various points in section 2.1 on apoptosis and necroptosis as per the reviewer’s suggestion.

Our main focus of the article (as reflected in the title) is the role of polyphenols in HCC treatment through modulation of autophagy. Hence, it would be extremely challenging to deviate from the spotlight. However, we have included the latest TKI drugs in section 2.7 (page 11, lines 454-474) as per the recommendation of the reviewer. Further, the role of protein kinases in regulating autophagy is also discussed in section 2 (page 6, lines 251-269). Multiple studies have shown that sorafenib (a TKI) induces autophagy, which is discussed in detail in the revised manuscript, e.g., page 10, lines 387-391; page 10, lines 430-432; page 11, lines 443-445; and page 11, lines 464-466.

Major concerns:

Comment 1:

In my opinion, the review should not focus on polyphenols, because although there are numerous in vitro studies that demonstrate their anti-tumor usefulness, there are hardly any in vivo studies or clinical studies that confirm this. However, there are drugs with potential or real usefulness against HCC, as demonstrated by in vivo studies and clinical trials, which can modulate autophagy, some of which appear in this review (section 6). TKIs should be the drugs on which the review should focus. In contrast, the polyphenol part can be included, much more summarized, but without being the focus of the drugs.

Response:

We thank the reviewer for the suggestion. We apologize that it is not possible to switch the focus of the review exclusively towards TKI. Our attempt is to summarize the collection of evidences on polyphenols modulating autophagy with respect to HCC. However, as per the reviewer's suggestion, TKI drugs have been included in the updated manuscript in section 2.7 (page 11, lines 454-474). Further, the role of protein kinase in regulating autophagy is also been included in section 2 (page 6, lines 251-269). We value the comment of the reviewer and will certainly consider the compilation of data on TKI towards HCC treatment in the future.

Comment 2:

In this sense, concerning polyphenols, a multitude of beneficial effects have been described as anti-tumor drugs. What is the importance of their modulating action on autophagy compared to their action on other described pathways?

Response:

We greatly appreciate this thought-provoking comment. Numerous experimental evidence exists to show that autophagy either plays a tumor suppressor function (or) use autophagy to comply with hypoxia and energy crisis that enable fast-growing and poorly vascularized tumors to survive and grow. Recent studies have concluded that activation of autophagy develops chemoresistance, therefore cytoprotective autophagy is the focus of several clinical trials. Nevertheless, if increased autophagy gives tumor resistance, its inhibition would lead to an increased response to the treatment.

A number of drugs that prevent autophagy initiation by ULK kinase inhibitors, nucleation by Vps34 inhibitors, elongation by ATG4 inhibitors, and lysosome fusion by chloroquine are currently recorded in preclinical and clinical trials. Enough evidence showed that sorafenib, a first-line HCC drug is able to promote autophagy-mediated cell death by overexpression of autophagy markers, including Beclin-1, by induction of ER stress and by suppression of the autophagy inhibitor mTORC1 and Akt.

Though polyphenols have anticancer effects with various molecular mechanisms, such as carcinogen-inactivating, antiproliferative, inducing cell cycle arrest, apoptotic and anti-angiogenic properties, autophagy modulation (inhibitors and inducers) is increasingly appreciated for the HCC therapy. Since drugs that can mediate autophagic cell death enhances the development of novel therapeutic strategies for HCC, eventually we have made great strides in studying the polyphenols influencing the autophagy process in HCC.

The importance of autophagy-related research can be fathomed from the ever-increasing number of publications (90 articles in the year 2000, 2050 in 2010 and 6700 articles in 2018; scopus data) and the fact that 2016 Nobel prize in Physiology or Medicine was awarded to Prof. Yoshinori Ohsumi for the research carried out on autophagy in yeast.

Comment 3:

Paragraph 2, and in particular the subsections on apoptosis, necroptosis and autophagy-regulating signaling pathways, are very extensive and are well-known concepts. They talk about cancer in general. Since the review focuses on HCC, the explanation in these sections should be oriented to HCC.

Response:

With regard to section 2 of the manuscript, as mentioned by the reviewer, we carefully reviewed the manuscript and excluded the molecular mechanism of known cell death. We have included the focus on HCC in section 2.1., in the revised manuscript (page 3, lines 131-135).

Since the current review emphasizes on autophagy, we are concerned that if we delete the sections on autophagy-regulating signaling pathways, then the following sections will not be clear to the reader. Therefore we would like to keep the autophagy-regulating signaling pathways in its current form.

Comment 4:

There are currently more TKI drugs that are approved against HCC, such as regorafenib, cabozantinib, and lenvatinib. Only sorafenib and its anti-tumor effects are mentioned in the review. Do these new TKIs effect on autophagy?

Response:

We had included sorafenib since extensive studies are available to show that sorafenib induced autophagy and increased lethality against HCC cells. However other TKIs targeting autophagy is also available as pointed out by the reviewer. Hence, the other TKIs have been included in the revised manuscript in section 2.7. (page 11, lines 454-474).

Minor concerns:

Comment 1:

Line 65. “Several anticancer drugs, such as vinblastine, teniposide, camptothecin, docetaxel, etoposide, and paclitaxel, have been identified from natural sources”.

Some of these drugs do not appear in natural sources but are semi-synthetic, obtained by modification of others that do have a natural origin.

Response:

The text has been amended accordingly (page 2, lines 63-66).

Comment 2:

Some typographical errors in the manuscript need to be corrected.

Response:

Typographical errors have been corrected.

Comment 3:

Doxorubicin is the preferred name over Adriamycin.

Response:

We agree with the reviewer. However, the drug has been cited as “Adriamycin” in the reference from which the content was obtained. Hence we have retained the name as adriamycin (Curcumin enhanced adriamycin-induced human liver-derived Hepatoma G2 cell death through activation of mitochondria-mediated apoptosis and autophagy; Reference No. 203).

On the other hand, the term “doxorubicin” has also been used in the manuscript in other places (page 11, line 449; page 13, line 554; and page 16, lines 687 and 688).

Comment 4:

Vinblastin is vinblastine (line 65).

Response:

We have made the correction as suggested (page 2, line 64).

Reviewer 2 Report

In this paper, the therapeutic effect for HCC of over-activated autophagy is reviewed. It is considered the possibility to induce type II PCD in cancer cells by autophagy and it is described the emerging evidences suggesting that the manipulation of autophagy could act as a potential tumor suppressor. Therefore, drugs altering the autophagic signaling could represent a new approach to the therapy of cancers resistant for current chemotherapeutic agents. Natural polyphenols upregulate tumor suppressors and autophagy by modulating canonical (Beclin-1-28 dependent) and non-canonical (Beclin-1-independent) signaling pathways. Furthermore, plant polyphenols target angiogenesis and metastasis in HCC by interfering with multiple intracellular signals. This review clarify the mechanisms of the HCC action of polyphenolic compounds through regulation of autophagy, a non-apoptotic mode of cell death.

This paper review has been well conceived and organized. I have no major objections. A minor point: the fig.s 1 and 3 contain some too small character’s that can be read with some difficulty.

Author Response

The authors of this manuscript express their sincere thanks to the reviewer for the critical assessment of our work. The authors have acted upon the recommendations of the reviewer which have resulted in a significant enhancement of the quality of this manuscript. All modifications incorporated in the manuscript are highlighted using red color font. A “point-by-point” response to the reviewer's comments is outlined below.

General comments:

In this paper, the therapeutic effect for HCC of over-activated autophagy is reviewed. It is considered the possibility to induce type II PCD in cancer cells by autophagy and it is described the emerging evidences suggesting that the manipulation of autophagy could act as a potential tumor suppressor. Therefore, drugs altering the autophagic signaling could represent a new approach to the therapy of cancers resistant for current chemotherapeutic agents. Natural polyphenols upregulate tumor suppressors and autophagy by modulating canonical (Beclin-1-28 dependent) and non-canonical (Beclin-1-independent) signaling pathways. Furthermore, plant polyphenols target angiogenesis and metastasis in HCC by interfering with multiple intracellular signals. This review clarify the mechanisms of the HCC action of polyphenolic compounds through regulation of autophagy, a non-apoptotic mode of cell death.

This paper review has been well conceived and organized. I have no major objections. A minor point: the figs 1 and 3 contain some too small character’s that can be read with some difficulty.

Response:

We are grateful to the reviewer for the encouraging comments regarding the quality of our manuscript. As per the reviewer's suggestion, the font size has been increased and orientation has been changed (for Figure 2) to improve readability.

Additionally,

  1. The reference list has been modified as we have added several new references and deleted many. Special attention is given to conform to the order of references and bibliographic style of the journal.
  2. The figures are modified with larger font size and changed orientation to improve readability.
  3. The entire manuscript has been thoroughly checked and edited to ensure uniform style, organization and quality.

Round 2

Reviewer 1 Report

The manuscript has been improved. The authors have followed the recommendations of the referee. Less relevant concepts from the review that were not related to autophagy have been reduced.

Only minor issues could finally be added to the text.
Although a sentence has been added at the end of paragraph "6-Clinical trials", I think it should be noted in the introduction section that there are numerous in vitro studies on the beneficial role of polyphenols in treating Hepatocellular Carcinoma, but there are hardly any in vivo studies and clinical trials. For this reason, it is worth investigating this aspect with these drugs alone or in combination for treating HCC.

Adriamycin is a trading name for doxorubicin. If you want to keep this name in the text because it was used in the reference, it could be indicated this way: adriamycin (doxorubicin).

Author Response

The authors of this manuscript express their sincere thanks to the reviewer for the re-evaluation of our work. The authors have acted upon the recommendations of the reviewer and once again revised the manuscript. All modifications incorporated in the manuscript are highlighted using blue color font. A “point-by-point” response to the reviewers’ comments is outlined below.

General comments:

The manuscript has been improved. The authors have followed the recommendations of the referee. Less relevant concepts from the review that were not related to autophagy have been reduced.

Response:

We are grateful to the reviewer for the insightful comments and appreciation of our revised manuscript.

Specific comments:

Comment 1:

Only minor issues could finally be added to the text.

Although a sentence has been added at the end of paragraph "6-Clinical trials", I think it should be noted in the introduction section that there are numerous in vitro studies on the beneficial role of polyphenols in treating Hepatocellular Carcinoma, but there are hardly any in vivo studies and clinical trials. For this reason, it is worth investigating this aspect with these drugs alone or in combination for treating HCC.

Response:

This is an excellent suggestion. We have added several sentences in the introduction section as recommended by the reviewer (page 3, lines 100-103).

Comment 2:

Adriamycin is a trading name for doxorubicin. If you want to keep this name in the text because it was used in the reference, it could be indicated this way: adriamycin (doxorubicin).

Response:

We agree and grateful to the reviewer's suggestion. We have indicated the details as recommended (page: 17, last line (in Table 2); page 18, line 729; and  page 21, line 812).

On behalf of my co-authors, I once again express my sincere thanks to the erudite reviewer for the valuable suggestions and constructive input to improve the quality of our manuscript.